# Parents’ Experiences of the First Year at Home with an Infant Born Extremely Preterm with and without Post-Discharge Intervention: Ambivalence, Loneliness, and Relationship Impact

**DOI:** 10.3390/ijerph17249326

**Published:** 2020-12-13

**Authors:** Erika Baraldi, Mara Westling Allodi, Ann-Charlotte Smedler, Björn Westrup, Kristina Löwing, Ulrika Ådén

**Affiliations:** 1Department of Special Education, Stockholm University, 106 91 Stockholm, Sweden; mara.allodi@specped.su.se; 2Department of Psychology, Stockholm University, 106 91 Stockholm, Sweden; acsr@psychology.su.se; 3Department of Women’s and Children’s Health, Karolinska Institutet, 171 77 Stockholm, Sweden; bjorn.westrup@ki.se (B.W.); kristina.lowing@ki.se (K.L.); ulrika.aden@ki.se (U.Å.); 4Karolinska University Hospital Functional Area Occupational Therapy and Physiotherapy, Allied Health Professionals Function, 171 76 Stockholm, Sweden; 5Karolinska University Hospital Neonatal Unit, 171 76 Stockholm, Sweden

**Keywords:** early intervention, follow-up, home visit program, infancy, parent-child interaction, parenthood, preterm infant, strengths-based approach, qualitative research

## Abstract

With increasing survival rates of children born extremely preterm (EPT), before gestational week 28, the post-discharge life of these families has gained significant research interest. Quantitative studies of parental experiences post-discharge have previously reported elevated levels depressive symptoms, posttraumatic stress-disorder and anxiety among the parents. The current investigation aims to qualitatively explore the situation for parents of children born EPT in Sweden during the first year at home. Semi-structured interviews were performed with 17 parents of 14 children born EPT; eight parents were from an early intervention group and nine parents from a group that received treatment as usual, with extended follow-up procedures. Three main themes were identified using a thematic analytic approach: child-related concerns, the inner state of the parent, and changed family dynamics. Parents in the intervention group also expressed themes related to the intervention, as a sense of security and knowledgeable interventionists. The results are discussed in relation to different concepts of health, parent–child interaction and attachment, and models of the recovery processes. In conclusion, parents describe the first year at home as a time of prolonged parental worries for the child as well as concerns regarding the parent’s own emotional state.

## 1. Introduction

Over 15 million children worldwide are born premature each year, causing a risk of long-term negative effects for the child and the family; this risk increases the earlier the infant is born and the less qualitative care they receive [1]. The World Health Organization (WHO) places particular importance on the continuation of care for children born preterm, from the hospital through to outpatient services and family and community care [2]. According to the Swedish Medical Birth Registry, extreme preterm (EPT) birth (i.e., being born before gestational week 28) affects about 400 children each year in Sweden [3]. EPT births require long hospital stays at a neonatal intensive care unit (NICU) often up to the date the infant was due and sometimes longer, but at least until the infant has reached 34 gestational weeks. Thus, staying in a neonatal unit for 2–6 months after birth is fundamental for the child’s survival. However, the long hospital stay is often a challenge for the parents, even when it is at infant- and family-centered developmental care NICUs. In Sweden, parents are generally expected to stay 24/7 as most units provide the necessary facilities. Parental experiences of NICU stay have been previously researched [4,5], where the amount of stress and disease is often related to the degree of infant- and family-centered developmental care (IFCDC) (e.g., Newborn Individualized Developmental Care and Assessment Program, NIDCAP) implemented in that specific ward [6,7,8]. Family-centered care may be summarized as a philosophy, as well as a practice of NICU care, where information is shared, parents are welcome to participate and collaborate with ward staff, and respect and dignity take precedence [9,10]. Additionally, the more specified the NIDCAP care, the better the understanding of the infant’s behavior. Medical and nursing care can, therefore, be tailored to apparent signs of stress in, for example, respiration, skin color, motor activity, and facial expressions [11,12]. Parents being offered NIDCAP have reported being more satisfied with the care offered [13] and witness fewer behavioral problems in their infants at the age of three [14], while staff experiences of NIDCAP have been generally positive [15]. The medical long-term effects are yet to be fully known, even though singular studies have reported positive results and no study has noted adverse effects [16,17]. 

### 1.1. Parental Mental Health Following Premature Birth

Quantitative studies of post-discharge parental experiences have described depressive symptoms, posttraumatic stress-disorder, and anxiety [18,19,20,21]. In children born EPT these psychiatric symptoms of the parent may affect the child’s outcomes at preschool age with regard to social and behavioral development [22,23,24]. Hence, understanding and caring as a parental experience affects the adult directly and the child indirectly. It seems that depressive symptoms among mothers with children born preterm may continue during the first whole year at home [25]. Some 20% of parents of children that were born very preterm still exhibit posttraumatic stress symptoms two years post-partum [26], while mothers of children born preterm who exhibit extreme or high amounts of distress, depression, or anxiety in the neonatal unit have an elevated risk of experiencing psychological distress one-year post-discharge [19]. Moreover, mothers with extreme or high amounts of distress, depression, or anxiety in the neonatal unit also have a more negative perception of their child’s abilities and strengths at one year of age [19]. 

Parental stress might be a broad and imprecisely defined concept of parental mental health. Studies point out that stress is common after a prematurely born infant, both from maternal [27] and paternal [28] perspectives. However, a meta-analysis of more than 3000 parents of preterm (<37 gestational weeks, GW) and low-birthweight (<2500 g) infants found only slightly more stress in parents of children born preterm compared to parents of children born at term age when measured during the first year at home, as well as later [29], suggesting that increased stress is not always present even among the parents of children born preterm. However, parents of children with lower birthweights and/or gestational ages at birth reported more stress than other parents of children born preterm, indicating that higher medical risks may increase parental stress. Levels of parental stress vary significantly between families with children born preterm [30] and it has been suggested that maternal stress relates more to role restriction, while paternal stress relates more to social isolation [31].

### 1.2. The Inner Parental Experience of Preterm Birth 

The inner feeling of becoming a parent prematurely has previously been studied using qualitative methods. The initial lack of control and proximity to the baby at the neonatal unit, from both psychological and physical perspectives, has been described as leading to feelings of disempowerment [32], grief [33], and mixed emotional experiences of sadness, guilt, anxiety, and worry [34] or even shock [35]. Some mothers describe their inner struggle with their bonding process to their child born EPT, and these negative feelings have also been reported as being accompanied by guilt and distress [36]. Some of the challenges at the NICU that parents have described include not only caring for the child and gathering as much information as possible, but also engaging with family, friends, and new NICU-acquaintances for perspective [37]. 

Aside from the biological definition of parenthood, there is an inner emotional transformation that occurs in parenthood, which has been described by parents of preterm children as a process with stages of alienation, responsibility, confidence and finally, familiarity [38]. This turns the limelight to the perspective of time, especially the importance of when in the process parents recalled their story of preterm parenthood. Feelings that arise during the NICU stay is one measuring point, as is the next step, which is the discharge process. The transition from being at the hospital to being at home with a child born preterm has been described as a challenge filled with joy and accompanied by worry, fatigue, social isolation, and misunderstanding, among other issues [39]. Discharge-planning when transitioning from the hospital ward to home care has previously been identified as an important factor for successful neonatal care and the parental experience [40], and is included in the recently published European Standards of Care for Newborn Health (ESCNH) [41]. About the time perspective, the emotional distress of parenting a critically ill newborn who has received care at a NICU does not seem to fade easily, and both grief and disempowerment, as well as the remembrance of family support and parental strengths, may persist for several years [33]. In the long term, there is also evidence (four years after giving birth) of greater personal growth among mothers that give birth preterm compared to mothers that give birth in the expected time [42], even though initial maternal mental health seems to be a mediating factor.

### 1.3. Preterm Birth Affecting Interaction and Attachment

Attachment [43,44,45] is the close relation a child forms with someone bigger, stronger, and wiser than them self (usually the parents), whom the child trusts for protection, support, and warmth. The warmth, sensitivity, and structure of the parent–child interaction is assumed to contribute to the degree of security in the attachment style that the child forms during the first year of life [43,44,45]. There is an ongoing scientific discussion on how preterm birth may affect the attachment processes, and some empirical work has been reported. When 117 children born very preterm or with very low birthweight (<1500 g) were tested with the Strange situation test [46] at two years of age, a lower frequency of secure attachment was found compared with full-term samples [47]. By contrast, and importantly, a systematic review found that differences in maternal interactions with children born preterm compared to children born at term age was the most evident during the first six months of life, and 5 out of 18 reviewed studies found equal or even higher interaction quality in the preterm mother-child dyads than the full-term dyads, when comparing free-play observations, still-face-interactions, feeding-situations, and other similar interactions [48]. Parents in a Danish study conducted in the NICU described hesitancy during early bonding with their child when the child had lower chances of survival [49], which is in line with previously mentioned findings that the first months are imperative when it comes to parent–child interaction in EPT-born dyads. Feelings of emptiness and self-perceived difficulties of estrangement have also been reported but mothers often describe how they eventually managed to form a bond with their child despite these difficulties [50]. 

Although the attachment process is not the focus of the present study, the parental perspective of the first year at home is certainly affected by how the parent–child interaction develops and the degree of security the child exhibits toward the parent.

### 1.4. Theoretical Approach and Previous Evidence of Home-Visiting Programs

Since toxic stress during the first years of life and prenatal stress on a child due to severe maternal stress in pregnancy may lead to persisting damage to the stress-response system of the growing brain and affect the individual physically and mentally into adulthood [51], intervening at this early stage is crucial for positive long-term development. Investments in early interventions have shown a much higher return than later efforts of helping disadvantaged children [52]. The focus of early interventions may be the infant, the parent, or the dyadic interaction between the two. Delivering the intervention in the actual home of the child often creates an opportunity to customize the service delivered to the specific family concerning, for example, better coaching of parents, more targeted individual strategies, and improved modeling from the interventionist [53]. To be effective, early intervention programs for infants with a high risk of atypical neurodevelopmental outcomes should target environmental enrichment, attuned parental responses, early communication skills, adjustments of sensory information for the child, and the promotion of motor development through scaffolding [54]. Considering the broad variation of kernels suggested, an integrated multi-professional approach to interventions would be beneficial. Specifically, when it comes to neurodevelopmental outcomes for children born very preterm, the parent–child synchrony has been suggested as the most important factor for positive outcomes at two years of age [55]. 

Even though the Stockholm Preterm Interaction-Based Intervention (SPIBI) is novel in a Swedish context, post-discharge home-visiting programs for children born preterm has been tested internationally. Some of the most influential programs are the Infant Behavioural Assessment and Intervention Program (IBAIP), the ToP programme [56,57,58], the modified Mother Infant Transaction Program (MITP) [59], and the Infant Health and Development Program (IHDP) [60], all of which aim to understand infants’ behavioral cues and respond accordingly. This cue-based thinking comes from Als’ synactive theory [11], where the autonomic, motor, state organization, attention, and self-regulation of the infant [61] are observed and the parental response is adjusted accordingly. The fundamental idea is that all behavioral subsystems are mutually interacting and simultaneously being affected by the infant environment. Stemming from Brazelton’s model of newborn full-term infants interacting with their mothers [62], the synactive model is also central to NIDCAP. Even if modern post-discharge programs for preterm infants share a theoretical background, their practical executions differ; some factors that may contribute to such executions include the intensity, implementation time, main focus, and the professionals involved. The MITP is administrated during the last week in hospital with two additional home-visits, focusing on sensitizing parents to baby cues and at the same time teaching them stimulating activities for their infants. The IBAIP specifically strengthens self-regulatory and co-regulatory behaviors. The Dutch trial of the IBAIP consists of 6–8 home visits from an infant physiotherapist before 6 months corrected age (CA). CA reflects the maturational stage of the child and corresponds to the age the infant would have been if it would have been born at term. 

The Dutch research team later developed the ToP intervention, which is now a part of standard care for children born very preterm in the Netherlands. The IHDP is a pre-existent intervention that has a longer scope of implementation (it lasts until three years of age) and also includes parental groups and a childcare program [56,57,58]. A comprehensive overview with steps for implementation of post-discharged responsive parenting programs was recently published in ESCNH [63].

### 1.5. Theoretical Approach of Health and Development

Given parents’ descriptions of their feelings and lived experiences of the first year at home with a child born EPT, the concept of health is an important notion. In 2006, the WHO defined health as a state of complete physical, mental, and social well-being [64], which may inspire states, institutions, and communities to continuously strive to improve the health of their citizens. However, the breadth of this definition also implies that quite a large proportion of human beings must be considered unhealthy in some aspect. Several other definitions of health have been discussed, including the absence of disease, the ability to realize vital goals, and a state of total well-being [65]. This article has adapted the theoretical approach that health is considered the ability to adapt and self-manage even in physical, social, and emotionally challenging times [66], since the complete medical recovery of the child is not always possible within the first few years following EPT birth [67], and neither is complete emotional recovery for the parents [25]. In order to reach health as defined by the WHO, efforts targeting the mental and social aspects of health in particular should be implemented, since preterm birth seems to affect family systems for several years [68]. 

In a bid to add more nuance to the health debate, this article puts forward different recovery processes as theoretical frameworks that facilitate understanding of the stories presented. A systematic review of 97 papers describing recovering processes from mental health found five processes including: connectedness, hope and optimism for the future, identity, meaning of life, and empowerment [69]. Even though the research question for this article is not mental health per se, the two concepts overlap in the parental stories of the first year at home. 

The theory of the developmental cascade [70] considers that progress or stress within one developmental level, through the numerous interactions and transactions between and within systems, may spread to other domains as well. This is certainly true for both negative and positive influences, and the basic idea of early interventions is, of course, to change childhood experiences, which may affect long-term outcomes for that person [70]. The developmental cascade theory is the reason why the parents were interviewed in this study, hypothesizing that what they report in their first year at home may also affect the wellbeing of their child in the long term. 

### 1.6. Aim

This qualitative study aims to understand how parents perceive their first year at home with a child born EPT. The specific research questions this paper will examine are:(I).How do parents of children born EPT describe the first year at home post-discharge?(II).How is participating in an interaction- and strength-based home-visiting program perceived by parents of children born EPT? 

We do not intend to evaluate the effects of the randomized controlled study (RCT) in the current study, since qualitative outcomes were not a predefined outcome [71].

## 2. Materials and Methods

### 2.1. Study Design

This study uses a qualitative descriptive approach to describe the lived experiences of parents with children born EPT during their first year at home. The interviews were performed when the child reached one-year corrected age, a time point chosen because it marked the end of the SPIBI early intervention program. The interviewer visited the home of the child to evaluate the first year at home as part of a larger SPIBI-study. The home visit also included filming of up to 45 min of parent-child playful interaction, collecting parental questionnaires concerning parental mental health and child development, as well as interviewing the parent. The family could choose the order of events, according to the state of the child in terms of alertness, hunger, and need for parental comfort. In most cases, the filming was executed first, since the length of the home visit risked tiring the child. 

### 2.2. Participants

A total of 17 parents of 14 children born EPT were interviewed; eight parents were from the intervention group and nine parents from the control group in SPIBI, as demonstrated in the flowchart in Figure 1 below. The target of the larger intervention-program is to recruit 130 families, and the 17 parents interviewed in this study are all included in the intervention study. As part of the intervention program, all 17 parents are part of an extended follow-up program, as described in the treatment as usual (TAU) condition section below. All participants received three extra discharge-preparing meetings with the recruiting psychologist and first author. Additionally, the eight parents from the intervention group received 10 home-visits by an interventionist. 

### 2.3. Treatment as Usual (TAU)

In Stockholm, all children born EPT are offered hospital-assisted home care visits as long as the child is tube-fed or in need of extra oxygen supply [72]. In addition to this, there is a high-risk follow-up program launched by the Swedish pediatric association in which, at term age, a standardized examination by a pediatrician is offered, followed by three months and one-year corrected age (CA) physiotherapist and pediatrician visits for motor and neurological progress assessments. This team meets the child again at 2 and 5.5 years CA, but at these two time-points a psychologist also assesses the child’s cognitive level and screens for communicative and behavioral problems. The pediatrician may refer the patient and the follow-up team will collaborate closely with a speech and language therapist, an occupational therapist, and a dietician. All participating families in this study received TAU. In addition to the TAU procedure, the recruitment process of SPIBI includes three coordinator visits, four baseline questionnaires for the parents, and one extra child physiotherapy assessment at three months CA. To clarify, the rather extensive recruitment and follow-up processes of SPIBI coupled with TAU is referred to as TAU^+^ in this article. 

### 2.4. The Intervention

The SPIBI consists of one pre-discharge hospital visit and nine additional home-visits during the first 12 months at home. The intervention is manualized and tailored to the Swedish context with surviving children from as early as 22 gestational weeks, 480 days of governmental funded parental leave per child, free public hospital care, and a general follow-up program for children that are at high-risk for neurodevelopmental problems, as described above. The SPIBI-program was developed in collaboration with the Dutch infant physiotherapeutic team that founded the ToP intervention [56,57,58]. The SPIBI is strength-based [73] and primarily concerned with parent-child interaction, understanding children’s cues, and encouraging the next developmental step. The aim is to reduce infant stress through parental support and to expand developmentally appropriate parent–child interactions for the well-being of both parents and children. There is evidence that increased parental self-efficacy through parental support is an important ingredient in family-centered practices during early childhood intervention [73]. Hence, parental feelings as well as behavior are the central target in the theory of change of the SPIBI-intervention and not the infant behavior per se. With increased parental well-being, less stress and reinforced parental responsive interaction, the intervention is hypothesized to affect the dyadic interplay, parental mental health and the infant development, see protocol for more information [71].

### 2.5. Two Semi-Structured Interview Guides

This article presents the data set from 14 semi-structured interviews [74]. Two semi-structured interview guides were created through the research team’s discussions concerning the information needed to qualitatively assess the SPIBI, which resulted in six questions for every participating parent and six additional questions for parents participating in the intervention group (IG) (see Table 1). The intention of both interview guides was to get an idea of how the first year at home was perceived, as well as to ask follow-up questions to the parents’ answers when applicable. The questions for the intervention group parents concerned the intervention and aimed to give a broader picture of parental perceptions compared to the client-satisfaction questionnaire CSQ-8 [75], which was distributed during the 12-month home-visit [71]. The general questions concerned parental perception of the first year at home, queries about the support received, and if parents required more or different types of support. The interview guides are quite different from each other, since answers from both groups could contribute to answering research question I, while answers exclusively from the IG could only answer research question II. 

### 2.6. Procedure

Two to four weeks prior to the one-year corrected age of the child, the first author, who is also the project coordinator of the SPIBI project, called one of the parents to schedule a time for the interview. The interviews were performed in the family home in 12 of the cases and by phone in two of the cases. Telecommunication was offered as an alternative due to the COVID-19 pandemic, which made home visits a potential risk for families with vulnerable infants. In three families, both parents took part in the interviews, in one family only the father took part, and in 10 families only the mother took part. One interview was conducted in English. 

The total interview time was 287 min, which meant that each interview lasted 20 min on average and the mean interview time did not differ between the two groups. The interviews ranged from 6 min to 39 min and took place from October 2019 to April 2020. 

### 2.7. Ethical Considerations

All subjects gave their informed consent for inclusion in the interview study before participating. The study was conducted in accordance with the Declaration of Helsinki, and the protocol was approved by the Regional Ethical Review Board in Stockholm. The initial application was approved in 2017 before the recruitment process started in September 2018 (ref. 2017/1596-31); two amendments relating to an extension of the follow-up measurements (ref. 2019/05169) and COVID-19 adjustments (ref. 2020/0117-73) were approved at a later stage. The study was published in ClinicalTrials.gov in October 2018 under the name SPIBI (NCT03714633).

Being interviewed about one of the most challenging events in your life may evoke intense feelings of grief, anger and worry. However, calmly sitting down with a psychologist and recalling the first year at home may offer an opportunity for mirroring and closure, as a part of the psychological process following every crisis experience. The parents were informed about the nature of the questions prior to participating in the interview and no parent has expressed negative effects after the interviews. 

### 2.8. Thematic Analysis

This article used thematic analysis (TA) as described in Braun and Clarke [76,77], which is a rather flexible method of identifying and analyzing patterns in qualitative material. The TA researcher played an active role in identifying and selecting the topics of interest and those that could both be facilitated and complicated based on the TA’s clinical experienced in the field. What was required for content to count as a theme was that it occurred in several data items, and whether it took up a considerable space in that specific interview or not. The participants could use different wording for the phenomena described, as long as the underlying meaning was the same and that it captured an important aspect of the first year at home with a child born EPT. Hence, the material was inductively analyzed, in a bid to find the latent meaning rather than the semantic wording of each theme. Highly sensitive material was either presented as overall themes, or when specifically cited, was read out loud by the actual informant for pre-publication approval. The TA-process can be broken down into six steps. Firstly, all interviews were transcribed verbatim by a research assistant. The original recordings were kept, in order for the first author as well as the two authors validating the themes to check for errors. No errors regarding the content in the transcriptions were detected. Each interview was named in chronological order by the date of the interview A-N. The first author familiarized themselves with the material by listening to the interviews and reading the 270 pages of transcription several times. Second, topics and specific citations from all the interviews were written down on an A2-paper, with one column for parents from the intervention group and one column for parents from the control group. Initial codes were formatted out of the topics and citations (i.e., “increased paternal worry” and “maternal depressive symptoms”). Third, the transcriptions were read through one more time, followed by grouping similar statements to form a mind map of candidate main themes and subthemes, regardless of group belonging (i.e., “different parental responses”). Transcriptions of the interviews were cut apart, attaching citations to the main theme that they belonged to (in some cases they were attached to more than one main theme). Fourth, these inductively generated candidate main themes from the original codes were then presented to the research-team, who gave feedback for revision and re-definition. The clarity of each citation, as well as the level of abstraction of each theme, was discussed. Since the transcriptions are full of medical terminology and abbreviations, the reader of the interviews must have NICU-experience to make sense of the data. Two of the authors who had not worked in the NICU, MWA and ACS, read the interview transcriptions to be able to validate and further fortify the themes. This was done to increase objectivity in the interpretation of the material. Finally, the defined themes were named and re-discussed by the research team and the report was completed. 

## 3. Results

The maternal age at the time of the interview ranged from 30 years to 44 (mean = 35.3) years of age. Out of the 14 interviewed families, six had a foreign background and 43% of the families consisted of at least one parent born outside of Sweden; one family had parents born in western Africa, four families had parents born in Asia, and one family had a parent born in southern Europe. The children were all 12 months of corrected age +/− two weeks at the time of the interview, but they were born from 22 + 6 to 27 + 6 (M = 25 + 6) gestational weeks. Four of them were girls and 10 were boys. 

Eventually, three main themes consisting of nine subthemes were identified as capturing the right balance of internal homogeneity and external heterogeneity to answer research question I. The main themes were child-related concerns, parental inner states, and family dynamics. There were surprisingly rich descriptions of development and worries concerning the child in the data material, despite the fact that the interview guides had only one question specifically relating to the child. The main theme of *child-related concerns* had subthemes of continued medical concerns, child regulation difficulties, as well as incomplete recovery when coming home. The main theme of *parental inner state* had the subthemes of loneliness (both solitariness and alienation), ambivalent feelings (e.g., the spectrum from relief to worry, from excitement to fear), and the process of keeping or letting go of a premature parental identity. Sometimes, the subthemes relating to the child gradually began to resemble subthemes relating to parental inner state, proving the interconnectedness of child- and parental-wellbeing during the infant year. The third main theme, which concerned *family dynamics*, was both related to siblings and marital issues (e.g., ranging from enriched differences in parental responses to more severe disagreements, or even estrangement). The main themes and subthemes are presented in Table 2 below. 

Participants from the intervention group evaluated their ten home-visits (addressing research question II), resulting in themes specifically related to the intervention: security, knowledgeable interventionist, and level of importance to the families, although not necessarily for both parents. 

### 3.1. The First Year at Home, Regardless of Group (Answering Research Question I)

The first year at home was described in a nuanced manner by both groups and the answers were presented in the following three main themes with attached subthemes. All the main themes were mentioned by a large proportion of the participating parents. 

#### 3.1.1. Main Theme 1: Child-Related Concerns

The first main theme consists of subthemes relating to the child and his/her medical state. Even though it had been a year since discharge at the time of the interviews, the vulnerable infant leaving the hospital seemed to be quite vivid in the parent’s minds. 

##### Subtheme: Continued Medical Concerns

The first weeks at the NICU are often described as a time characterized by a lot of technical devices, as well as medical and nursing interventions. The interviews show that such concerns continue post-discharge. As one parent concludes: “*Yeah, also when we came home, the first six months were like continuous hospital visits. So, it was like a non-stop hospital visit all the time. Then, we go to her eye, lung, heart, kidney doctor, everything*” (E, father from IG).

One child’s condition was referred to as initially unclear by the parents, as this quote exemplifies: “/…/and she should have surgery in March, we had so many hospital visits to [name of hospital], but then no, there was no surgery, and we had that information in April that [the cysts] had disappeared on their own” (F mother from TAU^+^). Uncertainty and unclear messages were identified as evidence of parental stress.

Most parents mentioned the use of technical devices during the initial period at home, such as apnea alarms, saturation meters, total parenteral nutrition, or extra oxygen supply. All parents also mentioned medical diagnoses, either as a fact, a worry, or in connection to a screening procedure. Some of the medical concerns discussed were gastroenterological issues, breathing issues, eye or brain injuries, as well as airway infections. For example, one parent described a tough re-hospitalization due to a common airway infection: “*One time we were hospitalized for five days, one week, the second time he became very ill. He received both high flow nasal canula and he received oxygen. I think the name was para influenza virus. No, he was in a very bad condition and it was awful, one week in the hospital and the memories returned*” (G, mother from IG). This shows that re-hospitalization may evoke more complicated reactions in the parent than what the medical diagnoses would imply. 

##### Subtheme: Child Regulation and Body Functions

Concerns regarding fundamental body functions, such as digesting food, sleeping, and breathing, may be persistent during the first weeks at home even with an infant who is born full term. However, these themes were still evident one year later in the parental stories. Parents also reflected upon their child’s sleeping behavior. One mother described how her child was awake at night for half a year rather than during the first three months as her friends with term-born infants had prepared her for: “*In the nights, it was hard. He was awake every night, from midnight or one, until five. Six months, it was very tough*” (A, mother from TAU^+^). In some cases, sleep deprivation was still present at one year corrected age, like this mother describes: “*/…/it is the sleep, sleep deprivation was my problem* [interviewer asks if she thinks this problem was more accentuated because of the EPT-birth, or if the issue would have been the same with a child born at term age] “*No, I cannot claim that, I have never squeezed out a baby and just gone home. I don’t have that experience, but I was damn tired, totally empty*” (D, mother from TAU^+^). Most often, stories of a child’s basic body functions and regulation related to eating, food, and the ability to gain weight. These could range from mild problems, as this father puts it: “*In the beginning he had difficulties eating, to accept the bottle. But it probably took just a couple of weeks, then it worked”* (C, father from TAU^+^) to more intense worries “*It’s important for these babies to keep gaining, to grow their lungs and all that stuff. So, we were a bit scared like, maybe they think of her like a normal baby, but for us it’s like she needs to gain weight. She cannot lose weight, so she’s not eating formula, what are we supposed to do?”* (E, mother from IG). Some parents described how worries regarding eating and weight still lingered one year later, despite their own understanding that these worries were no longer rational: “*even though he follows his curve, I am still totally obsessed with him gaining weight*” (I mother from IG). In this case, the parent understands that the rational thing would be to let go of the weight worries, although she does not manage to fully do so.

##### Subtheme: Incomplete Recovery When Coming Home 

Four children were re-hospitalized after discharge, two of them immediately after coming home. Both these families interpreted this as a medical misjudgment leading to untimely discharge and one of them was still very preoccupied with this one-year post-discharge, as can be seen in the following citation, describing a very difficult parental experience of severe apnea:
He had a cold at discharge, he received it in the neonatal unit, it’s written in the medical record that they suctioned his nose [of mucus], you know, some day before we went home. Everybody knew that he had a cold. But the cold got worse, that is the first thing I say to her [the neonatal visiting nurse] when she comes, “how are you?” and I answer “he has a bit of a cold”. And then I take him in my arms, and it is like holding your [pause] dead child. He is… completely lifeless. And totally grey, I still almost cannot talk about it. [Interviewer: even if it is one year ago?] Yes. So… eh… we try to get him back to life and then I leave him to her [the neonatal visiting nurse] and she makes him inhale…he is breathing a bit, and she says “no, you have to call 911”.(N, mother from TAU^+^)

Another parent describes incomplete recovery when coming home under less stressful circumstances, also resulting in a long re-hospitalization directly after coming home:
We were home for maybe one and a half days, and we noticed that [infant name] lost her muscular tone, which was a little longer, so we called and they transferred us to [name of hospital], where we came from [name of another hospital], so we called and they said we should go there, and we had to stay another week. They were a bit upset with [name of hospital] that they had sent us home so early, that it was completely, well, one doctor was very angry, thought it was too early. It is enormously stressful to come home and see that she did not cope with it. And, then, she received oxygen for one more week and they discussed oxygen at discharge.(L, mother from TAU^+^)

These rather dramatic parental stories of incomplete discharge processes emphasize the importance of discharge planning, as previously described in the literature. Even though parents are made aware that rehospitalization is common during the first year at home, the two families quoted above were not at all prepared for such acute situations at home so soon after discharge. Despite the fact that these rehospitalizations occurred one year prior to the interview, they were still very vivid in the parents’ minds and talking about them brought back negative emotions. Both families approved the publication of these specific quotes.

#### 3.1.2. Main Theme 2: Parental Inner State

The second main theme is the parental inner state and parental feelings, doubts, and definition of oneself. The subthemes are presented below.

##### Subtheme: Loneliness 

Loneliness was a theme evident in many of the interviews, from two different perspectives. First and foremost, the complex situation for the infant born EPT could not be easily understood by anyone who had not experienced it first-hand, leaving the primary caregivers with a feeling of being alone and lacking practical support. One parent gave voice to the feeling that relatives, even though they may be used to small children, did not know how to handle the special conditions of the child born EPT: “*It’s a, it’s a unique situation. Nobody else could do it, like: “I’ll just tube feed the kid”. No, it’s either me or his dad. So, we became very fastidious with him*” (D, mother from TAU^+^). Another parent spoke of the difficulty of teaching relatives how to greet the new family-member in an infection-safe manner at home:
So, just to meet other people, during wintertime when we came home, there were many infections around, so just telling people “you may not lift, not caress, you may not… keep away, you may look—but not touch. I think it is very hard for many people to understand, just that tiny part, you may look but do not touch. Because everybody says “oh, a baby”, [approaching body language] it is something imprinted in this…(H, mother from IG)

The fear of diseases, in combination with the common notion that it is hard for others to understand how severe a common infection could be for a sensitive child, is sometimes described as contributing to seclusion and loneliness, as one parent concluded “*we were very isolated in the beginning, in particular until March–April when the season of RS ended*” (I, mother from IG). 

Several parents described how they, even if their partners were with them, felt alone at home. This sometimes started at the hospital and intensified as the other parent went back to work.
Well my husband, he helps a lot, but when it comes to hospitals, he cannot handle…he thinks, he cannot handle the hospital environment for example. So, it has been tough for me, because I lived there for two and a half months. Many parents took turns, and that was ’what I missed the most. You have to take turns, because it’s just not possible.(G, mother from IG)

Another part of the loneliness relates to hospital discharge support. The regular hospital-assisted home-care nurse visits were described by parents from both groups as a supportive factor post-discharge. Two of the children did not receive that form of home-care support as they were fully breastfed and did not need any supplementary oxygen upon leaving the neonatal-ward. One of the families was in the intervention group, and described the situation like this:
I found it [the SPIBI-intervention] very rewarding and it has given me security, so I found it very good. We did not have any home care, for us it was quite abrupt because he was fully breastfed and had no oxygen supply when we came home around term age, so for us we went from meticulous monitoring, to practically not speaking. So, we were lucky to be a part of the study and have [interventionist name] coming to our home.(I, mother from IG)

This raises questions of the nature of hospital-assisted homecare in terms of providing a supportive role in psychosocial, medical, and feeding issues, rather than strictly medical hospital-assisted homecare nurse visits. The current system disadvantages breastfed children, as this does not ensure any additional neonatal nurse support. 

Another aspect of loneliness, better described as alienation, was mentioned by some parents. This may be interpreted as a form of loneliness in an existential sense in that nobody understood what the family had been through. One parent said that “*some people have understood, but I think very few people have understood in depth, what it is about, at [name of hospital] the doctors takes four hours at a time [regarding prognosis of the child] and I think that is hard for almost anyone to grasp, it was hard for me to understand before I*…[was in the situation]” (M, mother from IG). Even the staff at the neonatal unit were in some cases described as not understanding the challenges of preterm parenthood. One parent mentioned a discussion with a neonatologist in the aftermath of severe apnea in the home that the child managed to survive: “*No but, it was one of the doctors who, eh, tried to show some sort of co-understanding, but in the wrong way. “I understand how you feel”. Just “no, you don’t understand how I feel”. “But I can imagine”. “No not that either”. That is like, in a way, questioning, or a little/…/because you have a medical degree and know what to do, and I don’t”* (N, mother from TAU^+^). The interview made it clear that there was a need for professional sensitivity and high ethical standards in emotionally trying times for the family. There were examples of this need being met, as well as examples of the opposite. 

##### Subtheme: Ambivalence 

Ambivalent feelings were expressed regarding the discharge process, the attitude toward long-term consequences of preterm birth, and how best to handle different outcomes. The ambivalent feelings ranged from relief, freedom, and excitement to worry or fear. Some parents even expressed this ambivalence in the same sentence, by saying coming home was “*both fearful and exciting*” (K) or “*it was just nice. Of course, it was a bit worrisome with his apnea alarm and so on. But no, mostly just nice*” (C, father from TAU^+^) or “*it was really nice to come home, but at the same time I was really worried because he was so small during that time*” (A, mother from TAU^+^). More commonly, mixed feelings were expressed within the same interview, but not exactly in the same answer. The discharge process was commonly described in positive terms as a relief, a feeling of freedom, a time for a change, or a wonderful feeling. However, words like fear, worry, stress, and difficulties were also often used when describing this process. As one parent concluded, “*Sometimes, one just wants to scream, it was so tough*” (A, mother from TAU^+^). Another parent described the overwhelming experience to come home as follows:
It’s the first time he leaves the hospital, tastes air, sits in a chair or a baby car seat. Those were some tough hours. Yes. When he came home, we called the nurse and asked why his saturation was so low. Turns out, he had been sitting like this [shows crouched position] in the baby car seat and had not been getting enough air, so he was tired. So, when we arrived home, it took more time than we expected, maybe an hour./…/He was so tired, he could not even poop. And for us it was like “not today—not when we just arrived home”. (D, mother from TAU^+^)

Another part of the mixed emotional subtheme related to the parents’ feelings towards other parents in similar situations. They described how they had to face the fact that other parents whom also had children born preterm were suffering from more severe long-term consequences. The hardships of handling the different outcomes of children born EPT were described like this by one parent “*[name of child] is quite well, it’s not comparable to the other two babies at all, I think. It could be a large grief for them [e.g., that their child had more complications of preterm birth than mine]*/…/” (L, mother from TAU^+^). Another parent, who in general was very positive about her first year at home, still described guilt when comparing her experience to another parent’s experience:
But, when that day came [the one-year corrected age birthday], then one felt a bit guilty. Because when we speak with this group, they wrote, they said that they lost one of the twins, so it has been tough for her, he died in the womb. And the other, I think she had some anxiety before, so, they have felt really bad.(G, mother from IG)

On the other hand, some parents described sadness related to the fact that their child did not develop as the children born at term age in the circle of acquaintances, as this utterance exemplifies: “*And then when you meet the children and see that they are like…One child was walking and had the first preschool day last Monday, and that child walked with a backpack and everything. One meets other children who are… one may become a bit… well it can be tough*” (H, mother from IG). This ambivalence can take the form of guilt regarding the development of one’s own child in comparison to other children born EPT, sadness for other families combined with relief for one’s own child, worry for one’s child’s development in combination with faith that it would end well; this subtheme was evident in some form or another in most of the interviews analyzed. 

##### Subtheme: Preterm Parental Identity

Several parents think about the identity of the child as an EPT-born child, as well as their own identity as a parent of an EPT-born child, often highlighted by participation in the parental group on Facebook: “Extremprematur—föräldragrupp” (with nearly 1400 members who are all parents of children born EPT) [78]. This subtheme could also be interpreted as an urge for social support and networking with individuals who truly understand the struggle, as they have had similar experiences themselves. This could also be linked to the aforementioned subtheme of loneliness, as a community could act as a remedy for solitariness. One mother reasoned about her tendency to still cling to the prematurity identity:
So, it’s a bit scary, like it feels like we have now distanced ourselves from that [the extreme preterm birth] then maybe you should not stick in this [Facebook group] flow of the process. So, the thought has struck me if I should sort of leave that group, because we are trying to let it go now. Because you always hear… there are a lot of problems in these kinds of groups as well, how people air their concerns./…/Or it’s something that makes me hang in there [in the Facebook group], a little. Still. For some reason.(F, mother from TAU^+^)

Several parents mentioned the Facebook-community as a source of knowledge and experience from a parental perspective that they missed in regular hospital care. One mother described the help she received from another parent who had been in a similar situation before on social media: “*She explained that many things will happen, the CVK [central venous catheter] will break, the child pulls it and, there are several ways they may break. Ehh, sepsis, it will, it can happen, so I could prepare, okay, if it happens, I just go in* [to the hospital]” (J, mother from TAU^+^). Such clear instructions and frank preparations for the first year at home were not described as part of the information provided during the hospital discharge process. 

Others described how at this point they had left the neonatal identity behind, as conveyed by this parent:
I cannot relate her to her early birth any longer. But when I see the photos in my phone, that I took [in the hospital] at that time, then I feel “has that happened to me?” One forgets so fast and that is good, in one way, I think, because it has been OK, she is here and growing. Why should I go back and dwell upon what has been when there is no point?(K, mother from TAU^+^)

The parents expressing this point of view (i.e., that the preterm birth belonged to the past), all had children who did not have any known disabilities at the time of the interview. They could focus on everyday life with a one-year old child at home, and found no point in looking to the past. 

#### 3.1.3. Main Theme 3: Changed Family Dynamics

The last main theme does not relate to any special profile, but more to the transactions, or relations between individuals. When a child born EPT enters a family, the dynamic tends to change.

##### Subtheme: The Parental Dyad

A large portion of the interviews were clearly related to the new family member and the parents’ relation to that child. Although it was less evident, most interviews also contained descriptions of how the preterm birth brought changes to other close relationships during the first year at home, as the parents exhibited different reactions. Differences in parental reactions within the family were reported in 10 of 14 interviews; more worry, sadness, and complex feelings were reported by the mother in five of the cases, by the father in three of the families, and other differences were noted in two of the families. 

The differing parental responses sometimes meant that the father was more worried, as described by this mother: “I felt confident that he will be fine. He does not have any apneas any more. But the father… eh… did not feel as safe, but would have preferred to sit by the crib to see and watch over him all the time” (I, mother from IG), or another mother “he [the father] was a bit more…skeptical, that it was too early, that it would not end well, and he was scared. As for me, that was not the feeling. No, it was OK, I felt ready” (K, mother of TAU^+^). But it could also be the other way around as one mother said, “I think he [the father] was, as he said before, he was more positive from the beginning” (E, mother from IG). Some mothers, but no fathers, described how they were stressed to the point of being on the verge of depression. One mother described her mental health development during the first year at home like this:
Yes, but then he went back to work in March, yeah, so he stayed home a bit longer. Uhm, but then I felt… I felt quite bad mentally during spring, I felt extremely stressed, like I was approaching fatigue depression, everything stressed me out, just visiting the well-baby clinic [BVC] was stressful. I don’t know if it was, that so many things that caught up from the past time, or the shock of becoming parent of two children, which was much tougher than I ever imagined. Or if it was interconnected.(F, mother of TAU^+^)

Some partners described how their different views of the preterm birth had affected their marriage. One couple had separated during the first year at home, an event that the interviewed mother related to the sojourn at the NICU. At the same time, three couples pointed out that they would never have managed the first year at home if not for the other parent (one because of severe somatic illness in the mother, one because of the instant need to go back to work due to the child’s predicament pushing them to the verge of emotional burn-out, and one couple said that taking turns caring for the child saved them from mental breakdown). Below, three different couples tell their stories of how the first year at home affected their relationship, to a larger or lesser extent.
And then when we change TPN [total parenteral nutrition], from the beginning we said that I changed one time, then my husband changed the next. But his, you know, men are not so careful as women are, or not so [pause] skilled as women. So, I just told him “you have to do it like this” and he got very angry, because he thought I was saying I was more capable than he was. It’s not like I don’t believe in him, even if that is what he thinks. But, the first period, it was about how we fed him, how we gave him medicine, replaced the tube. It was a bit tense at times, during that period.(J, mother from TAU^+^)
It is incredibly stressful to come home and like try to handle that you are coming home with a newborn, but also somebody that fragile, and then I was eh… more or less all alone in this situation from discharge/…/At the hospital, that was at [name of hospital], we were, there were so many staff members that came to us and were fascinated by how nicely we handled it together and that was my experience too. But then, as I said before, when we came home… then… well it was a personality change, absolutely. (L, mother from TAU^+^)
Well, we, it felt like we are persons who “We will be fine, we’ll be fine, we do not need anybody”. But this time I felt more like “no, we will not be fine—we need help” and there is nothing wrong in asking for help. (M, mother from IG)

##### Subtheme: Thoughts Relating to Siblings

Another change in family dynamic described by all second- or third-time parents, was the situation for the siblings. In some cases, being left at home while the parents spent several weeks in hospital rendered feelings of guilt for the newborn’s siblings. Previous experience of child-rearing was also discussed, with first-time parents mentioning how hard it must be to have an older child at home to simultaneously care for. Second/third-time parents discussed how much easier parenthood was for them, having had prior experience. This highlights that children are not born in a vacuum, but rather are introduced to a family, a surrounding environment, and a context which profoundly affects their life and development. As one experienced parent concluded *“I’ve had two children before, so I was not so scared*” (K, mother from TAU^+^), other parents focused on the hardships of having several children at home: “*but then it was that we had the older sister at home [not in preschool] for a while, it was very demanding. So, there you, or me as a mother felt that I lost this precious infant year*” (F, mother from TAU^+^) and “*the other children have been very affected/…/And we have explained, “no we cannot have a birthday party, because he has just come home from the neonatal unit, like last year*” (G, mother from IG). The experience of having other children was also discussed in positive terms however, as one mother complimented the care provided for them in hospital “*The siblings always felt engaged and they knew much more [about preterm birth] than they would have. They were always welcome. So, they felt that they also lived in the hospital* [laughter]” (I, mother from IG).

##### Subtheme: Intergenerational Support

Half of the families mentioned relatives as a steady source of support and in some cases a grandmother had moved into the family home for a period of time, as this mother recalled:
So, in that way we have had the support that somebody, if I regained lost sleep, the first week my mother lived with us and then I could sleep while he was awake with my mother. So, because she is not updated in his [medical state], but just to cook, clean, and everything. And then the second week, his [the father’s] mother came and lived with us and did the same thing.(D mother from TAU^+^)

Grandparents not living with the family could still be a source of support. One mother described how “It was a giant machinery of family and friends who really supported me and said you have to talk to, even though you feel that you can handle this here and now, you have to talk to a psychologist” (L, mother of TAU^+^). Support from relatives or extended family came in several forms ranging from taking the child for a walk in the pram and arranging grocery shopping for the family, to moving in with the family for a period of time. The much-appreciated solution of grandmothers living with the family for a period of time was exclusively seen in families where the grandmother was born outside of Sweden. 

### 3.2. Specific Themes Connected to the Intervention, Answering Research Question II

In order to answer research question II, the six interviews from the intervention group were also analyzed separately. The stories from parents who participated in the intervention group of the SPIBI post-discharge program during the first year at home contained additional themes of security, the value of a knowledgeable interventionist giving specific advice, and the value of an important, but not always necessary, supportive service. Amongst several of the families, the intervention was perceived as consisting of three parts: psychological parental support, physiotherapeutic skills, and eating support. One parent described it as follows:
It was in three parts, so in the beginning I felt that [name of interventionist] was like a psychologist almost, just so you feel that…You have had psychological support/…/And then it is a bit like a physiotherapist, she has coached me a lot, like “he will soon start with this, so do this”/…/and then the food, when one has been forever worried, that we could discuss that a bit.(I, mother from IG)

#### 3.2.1. Intervention Group (IG)-Theme 1: Security 

All interviewed parents talked about the sense of security that participating in the intervention brought them. This could be expressed as a lack of fear: “*In the beginning we were like, a bit scared and that time I think that the intervention was helpful, because we could ask all the questions*” (E, mother from IG). Others described trust in the infant’s capacity “*Yes, regarding that he was preterm born and we thought, he is, you know, maybe different than his brother, he may need more help but [name of interventionist] was here a lot and it helped, it felt more secure, she was just like* “*but let him be, he will be fine*” (M, mother from IG). In challenging times, the intervention was described as extra, well-needed support: “*It felt safe to have someone, even if we maybe did not always need her, it has felt secure too, because, at the time that I told you about when I did not feel so well, it was so nice to have extra support*” (B, mother from IG). The notion that the intervention was appreciated, but not always needed, perhaps indicates a need for some autonomy following such a long and highly dependent hospital stay. 

The security theme’s importance is also highlighted by its absence amongst parents receiving the extended follow-up program within the treatment as usual (TAU^+^). When asked how the hospital could support parents post-discharge, one mother from the TAU^+^ group answered:
More home-visits maybe. Sitting talking, like we do now. How it works at home, if it is something we need… But when we came home, it was all over, it was just me going there all the time because of the [child’s] constipation. Nobody came. I asked somebody there when I was worried that it will be a tough situation at home. They said ”no, you don’t need home care because he does not have a tube or anything”/…/but still, he was not very…very strong/…/but you know, the worry is stuck here [points to her heart].(A mother from TAU^+^)

Even though the theme is security, it may also entail a need for psychological aid, or what has been referred to as a ‘containing function’. The interventionist was described as somebody who was reliable and safe, highlighting the need for continuity, but who also understood the special conditions required by many of the EPT children and their parents (see theme below). The combination of these elements can be described as the interventionist serving a holding function for one year, with the interview marking the end of that era. Some parents described the sadness of ending the intervention: “*it is almost a bit sad, it is a bit like the end of the circle*” (B mother from IG), and another mother solemnly concluded “*so one has to let even you go*” (M mother from IG). Despite the staff in the child’s life changing over time, the SPIBI interventionist remained constant throughout the first year (until the infant reached one year corrected age).

#### 3.2.2. IG-Theme 2: Knowledgeable Interventionist 

Parents mentioned the value of having somebody give them perspective regarding their child’s development. One parent spoke of her sudden realization of all the things her child actually understood, surprised at how far he had come in his language development and passive understanding:
I think it gave me quite a lot, regarding that, eh, I learned to think in another way, it opened my eyes, I had an “Aha-moment”, for example things that you take for granted, you think he does not understand, but all of a sudden, “oops, he understands”. (M, mother from IG)

Overall, parents valued how knowledgeable their interventionist was, regardless of which interventionist they had. These three parents expressed this in different ways:
It has not always been us posing questions, more that she mentions or says this is common amongst preterm born, or extremely preterm born … and maybe you have not thought of that at the time, but then when you compare to other children, or what shall I say, at some point, then you are not like “but why does my child don’t do that”, because she has prevented that worry. (H, mother from IG)
It was very fun to be able to participate, really, when [name of interventionist] came and gave tips and suggested what to do, things I did not think of with my previous children, “aha, that is why they do that, it is for this reason.(G, mother from IG)
It was good, because as [name of child] is our first child, we did not know much about parenting, so when [name of interventionist] came and said “she’s going to do this thing in a few days, a few weeks” it was helpful for us.(E, father from IG)

Many parents highlighted the practical parts of the intervention as being especially helpful, with interventionists often being able to find suitable tips and advice on activities to, for example, strengthen the infant’s motor development. This could be expressed as support for stability “*She gave suggestions of how he could sit, with a towel around because they develop a bit later sometimes regarding their motor skills, then children normally do. So, it has helped a lot*” (G, mother from IG), or more like playfulness “*So she gave some tips and tricks, how to train his neck and how he should play on his left side, because he tended to play on the right side, little things*” (B, mother from IG).

#### 3.2.3. IG-Theme 3: Important, but Not Necessary 

Some, but certainly not all, families described their children as lucky to not have had severe complications during their first year at home. They still described the intervention as helpful, but not critically important. As one parent vigorously put it: “*It has felt like a luxury sometimes, because we have not always needed that care at all times/…/because he has not had so serious problems actually*” (H, mother from IG). This shows that even in this group of families, with children who needed advanced medical assistance at birth, the parental need for further support and help at one year of corrected age varied greatly. This theme will be discussed in relation to their socioeconomic background, which, to some extent, co-varied with the autonomous utterances that the home-visits were a welcome luxury, but also in relation to a multilayered approach to follow-up care. 

## 4. Discussion

Three main themes of parental experiences, answering research question I, were identified using a thematic analytic approach: child-related concerns, inner state of the parent, and changed family dynamics. Parents of the intervention group also expressed themes related to the intervention, as a sense of security and knowledgeable interventionists, answering research question II. 

First, the results of research question I (i.e., parental inner state, child-related concerns, and changed family dynamics) will be discussed. The subthemes of the parental inner state are in line with previous reports concerning depressive feelings [18,19,20,21] and stress [27,28]. Even though Sweden has active and advanced NICUs and family-centered developmental care [7], feelings of emptiness and isolation are not uncommon post-discharge. This points to a need for multi-professional services, where a more integrated perspective of the child also includes parental mental health aspects. Furthermore, feelings of loneliness are quite common and forms another subtheme. In this regard, cultural aspects of the loneliness subtheme should not be neglected. Some of the families from foreign backgrounds described how a grandmother came to stay with them for a few months even if the grandmother was not a resident of Sweden, while another family stayed with the grandmother for a period of time, which was described as being very helpful. The grandmothers living with the families were described as quite involved in the infant’s development, as well as any infant difficulties. A recent study on the intergenerational perspective of families with a child diagnosed with autism spectrum disorder in Sweden suggested that grandparents expressed a strong interest in receiving more information about the grandchild’s special needs and how to support the family in difficult times [79]. Since the grandparents were described as a source of help and support by some parents, it would be interesting for further research to investigate how they themselves perceive the situation and whether they themselves require aid. Notably, intergenerational custodial solutions were not evident in any of the fully Swedish families interviewed. However, in an individualistic society [80], social media may, to some extent, fill the role of the much-needed post-discharge companionship supporting their identities as parents of an infant born EPT. As a community based on the internet, social media users do not have the scientific background or the medical/nursing degree sometimes needed for optimal support. However, the support from other parents and from the Facebook group was described by some as more candid and preparational than the hospital-assisted home-care. 

The mixed feelings that are often exhibited in qualitative studies of parents with children born preterm [34] were also evident in this study, as described in the ambivalence subtheme. Of course, becoming a parent to an initially very fragile child may be an unsettling experience. The question is whether this overwhelming feeling could be relieved by more comprehensive and careful discharge planning. Since persisting medical concerns and even incomplete recovery when returning home (resulting in acute re-hospitalization) were subthemes in the material, criticism regarding the discharge process is understandable. Questions regarding parental preparation, not only concerning medical issues but also the psychological aspects of coming home, were raised by the participants of this study. Discussion concerning discharge planning is not new [40] and the transition from hospital to home has previously been described as a challenge and a potentially socially disruptive experience [39]. This raises the question of whether discharge planning would benefit from a more holistic approach to coming home. Swedish data shows a re-admission rate of 5.2% among children treated in hospital-assisted neonatal home-care, and being born EPT was a risk factor for re-admission [72]. In this material, 2 of the 14 children were readmitted during the first week at home, and their parental stories includes extreme stress and very vivid negative emotions one year following the event. Counterintuitively, the families with medically assessed low-risk infants (e.g., fully breastfed and no need for extra oxygen) were among those expressing the highest need for additional support. This highlights the fact that a medically stable child does not necessarily translate to a psychologically stable situation at home. It is not uncommon that medical evaluations are mistakenly equated with psychological assessments in neonatal wards. Acknowledging the distinction and need for both is crucial for the parent’s wellbeing. A definition of health that includes self-management, even in times of physical, social, and emotional challenges, [66] seems to be a necessity in this context. As previously mentioned, WHO defined health as a state of complete physical, mental, and social well-being [64]. This definition points to the importance of not only including the somatic medical state of the child, but also parents’ mental health when assessing the health situation of families with a child born EPT. 

Previous research clearly shows that leaving the womb three to four months prior to the expected due date has an effect upon both the child [67] and the parent [18,19,20,21,27,28], as well as the relationship between the two [47]. Our qualitative study revealed the subtheme of changed family dynamics, which illustrates that preterm birth induces a domino effect, influencing several other important close relations as well. Most frequently cited were the marital relation and the sibling situation. EPT birth, a long hospital stay leading to separation between family members and uncertainty about the future, is unsettling for everyone involved. The arrival of a new family member always influences the delicate family balance to some extent, but the added pressures that come with EPT birth may make it even more unstable. From the main interview themes, it seems that when the survival of the new-born child is at stake, other relations are re-evaluated as well. Close relations are often described as a psychological buffer in times of crises [81], though this was seldom mentioned in the interviews. One parent described taking turns at the hospital as a steady base for the post-discharge situation at home, while experiences of relational instability were much more common. Differences in parental responses to the EPT birth was reported in 10 of the 14 families interviewed; sometimes the mother and sometimes the father were described as the more worried parent. In Sweden, fathers receive 10 governmental paid days off from work following the birth of a child, but when the child is born EPT, it is common that the father stays at the hospital with the mother for the whole NICU stay and beyond, often resulting in three months off from work, which is financially possible thanks to Sweden’s “paid care for a sick child-policy”. Being a father engaged in child development sometimes also implied increased worry in some of the stories. This raises the question of whether staying close to the newborn child may increase the fathers’ feeling of worry, despite also potentially improving their engagement in the child’s upbringing. Later analyses of data obtained from questionnaires concerning parental resilience and mental health may contribute information to this matter. 

The first year at home may be interpreted as a year of recovery. Recovery processes from mental health problems have been described as including connectedness and identity, amongst other themes [69]. Even though becoming a parent to a child born EPT does not necessary lead to mental health complications, the recovery processes from extreme preterm parenthood and mental health issues, respectively, seem to resemble each other in some perspectives. The emphasis on letting go of the identity as a parent of a child born EPT was something several parents talked about as the child turned one-year corrected age. The comfort of being a part of a community, further highlighted by access to a specialized group on social media, was not easily abandoned. The action of engaging in new NICU-acquaintances has previously been described as a coping-strategy at the NICU [37], and this study suggests that the process continues after discharge. Social media appeared to offer a unique arena for connectedness with other families in the same situation. This raises the question of how long this identity is helpful for the parents and when, if at any time, primarily seeing oneself as a regular parent becomes beneficial? One should be aware that parental processes of children born preterm change over time [38], and what is considered an important theme one-year post-discharge may not be important several years later [33]. 

Answering research question II, the interviews revealed that three components—psychological parental support, physiotherapeutic skills, and eating support—were perceived as cornerstones of interventions. These were certainly a part of the training received in the SPIBI, but had not been conceived by the research team as three distinct pillars of intervention. Nevertheless, they were mentioned as such by several parents. Their salience is in line with the literature on family resilience [82], according to which individual factors of internal locus of control, self-efficacy, and effective coping-skills represent the psychological parental support; increased skills, training, and an individually tailored stimulating environment represent the physiotherapeutic skills; and supportive parent-child interaction is the eating support. Family resilience is a complex interaction between inhibiting the impact of risk factors and strengthening protective factors [82]. In the SPIBI population, there are several risk factors present, not only medical fragility and early parent–child separation but also socioeconomic challenges. Hence, if the SPIBI intervention indeed supports factors relating to family resilience, this may contribute to a more stable situation for the children born EPT. 

Attachment theory [43,44,45] highlights that the infant is not a passive and weak-willed organism being thrown into this world, but rather is born with the ability to purposefully interact socially and physically with its surroundings. This is commonly referred to as “the competent child” and research suggests that a parent’s inner representation of their infant could be related to the attachment style classification in the strange situation test, where the child’s reactions to parental separation is carefully assessed at one year of age [83]. NICU care is an acute medical specialty, requiring constant focus even on small changes in infant cues and their medical condition. It is fair to say that NICU care targets illness, immaturity, and problems out of medical necessity, which at times may contradict what is needed for that specific child’s psychological growth. Advanced hospital care is quite occupied with vulnerability and curing illness, while family-centered developmental care [9,10] and NIDCAP [11,12] in the NICU act as a resistant force focusing on the capability and resilience of both the newborn and the parents. However, parental skills and the inner representation of the child need to be reinforced and supported even after discharge. There are some statements from the interviews, noted in the IG-theme of security, which show that the interventionists helped parents focus on what the child actually manages to do, in order to induce hope and trust in the child’s capacity rather than focusing on what he or she is too sensitive to master yet; this forms the foundation of a strengths-based approach [73]. Psychology as a science has been criticized for being ill-equipped at preventing problems, as well as diseases, and a positive psychology approach focusing on resilience and salutogenic strength-based interventions can counter this [81]. The idea of SPIBI intervention is to strengthen the signs of positive interaction that already exist, rather than pointing to eventual weaknesses or problems, which is in line with a positive psychology stance. In later years, not least within positive psychology, research on developmental cascade effects has received increasing attention and empirical support [70]. The cascade refers to the fact that one event or component may serve as the starting point for a series of events, interactions, and transactions that drive development. In more abstract terms, the cascade can be understood as the accumulation of consequences for development in developing systems, including in terms of the spreading of effects. The rationale for SPIBI fits well with this conceptual framework, even if the theory of change is founded on increasing parental responsive interaction with the child and reducing stress, this may be seen as a cascade of positive transformation in parental behavior, knowledge and feelings, affecting the infant’s development and interactive behavior as a cascade effect. 

Parental answers regarding the intervention were generally positive, short, and concise. Some parents wished less frequent home visits in the beginning of the year, and more intense treatment by the end of the first year when motor development had gained momentum. Some families did not consider the logbook helpful, while one family thought that more written material, especially an overview of the themes and visits, would be useful. All these suggestions will be considered when further developing SPIBI intervention (once the current RCT is completed). 

All the intervention families who were interviewed expressed satisfaction with the intervention, even though they varied in how much they felt it was necessary. This highlights that there may be reasons to tailor the service to each specific family [53]. This result is in line with previously published best practice guidelines recommendation of a multi-layered approach to supporting parents of infants born preterm in the NICU [84]. During the interviews, some parents concluded that the EPT birth did not have any lasting consequences for their child and, hence, they could leave the preterm parental identity behind, turning their attention to everyday life. This feeling may or may not be long-lasting should the parents be re-interviewed some years later. Many neurodevelopmental disorders are commonly discovered later, during the child’s preschool-years [67,85].

An important methodological discussion concerns the representativeness of the interviewed parents for families with a child born EPT in Sweden. In Sweden, 19.6% of the population were born outside of Sweden as of 31 December 2019, and in Stockholm County 36% of the population have foreign backgrounds [86,87]. The Swedish population-based follow-up study by children born EPT in 2004–2007 revealed that 18.1% of parents had a non-Nordic land of origin [85]. In this study, 43% of the families had at least one parent who was born outside of Sweden, which is quite a large proportion. Since one exclusion criteria of SPIBI is families where none of the parents speak Swedish or English [71], some of the families who recently immigrated to Sweden were initially excluded. Previous research has suggested that migrating to Sweden may increase the risk of giving birth preterm during the first year [88], a result partly confirmed by international data [89], even though clear conclusions are difficult to draw since the definition of migration is not standardized. Once the RCT is analyzed, it remains to be seen if families with a history of migration in their recent past are over-represented in our EPT study group. 

Thanks to the broad representation of parents described above, the recruitment base from four different hospitals in Stockholm and the spread of gestational weeks at birth of the children, the results of this study may with caution be generalized to the parents of children born EPT in Sweden. However, the results are not considered transferable to parents of premature born children in Sweden in general. Several of the themes from the interviews were rather closely related to the amount of time spent in the hospital, where the level of frustration, worry and disappointment increases with the time before discharge. In addition to this it cannot be ruled out that some of the themes also mirror the medical state of the child with possible disabilities, a risk increasing with lower gestational age at birth. Due to the different standards of neonatal care in different parts of the world, transnational generalizations are not recommended. 

Another methodological aspect is to discuss concerns over how and by whom the themes were found. For transparency, it is worth noting that the main author/interviewer had six years of experience in psychological clinical work in neonatal units, of which four years were in combination with research in the field as well as project-based work on a national level concerning guidelines for psychosocial work in neonatal units. This pre-study knowledge contributed to the challenge of listening to the interviews with a clear and somewhat open mindset, perceiving the nuances of the stories and performing inductive analyses. At the same time, the years of clinical work in the field helped gain the parents’ trust during interviews, as well as helping to understand the nuances in the material, which was full of medical abbreviations. A more indisputable limitation was that most interviews were frequently interrupted by a one-year old child born EPT, who was present in all interviews but two. 

The strength of this study came from the multi-professional research team that interpreted the data. However, rich experience may also act as a limitation, since several years of neonatal work may inadvertently guide the analytical work of the data in a particular way. 

## 5. Conclusions

The first year at home with an infant born EPT is shown to be a time of prolonged parental concern for the child as well as for one’s own emotional state. Furthermore, having a child born EPT affects several other relations as well. A post-discharge program focusing on psychological parental aspects of preterm birth, early motor development, and parent–child interaction was welcomed and appreciated, following advanced NICU care for children born EPT and their caregivers. Further research is needed to fully understand the impact of an early post-discharge program for children born EPT, but also to understand the paternal as well as intergenerational perspective of the first year at home with a child born EPT versus a child born at term age. 

## Figures and Tables

**Figure 1 ijerph-17-09326-f001:**
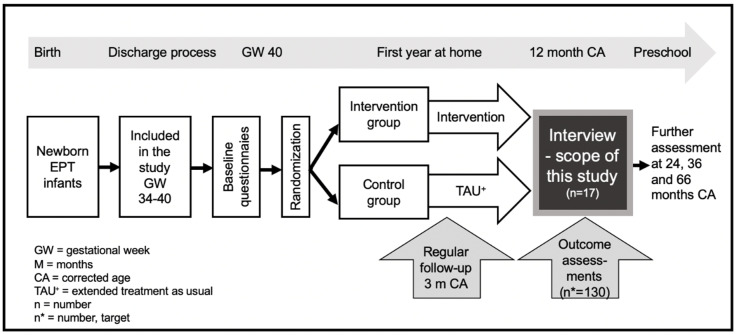
Flowchart of Stockholm Preterm Interaction-Based Intervention (SPIBI).

**Table 1 ijerph-17-09326-t001:** Semi-structured interview guides: for both groups and for intervention group.

Interview Guide	Additional (IG) Interview Guide
How did you perceive coming home from the neonatal unit? (a) What support did you receive from the hospital and care system? How did you perceive it? (b) Has any other support been offered to you from your local community, society in general, social media, parental groups, friends, relatives etc.? If yes, what kind of support and by whom?	How did you perceive participating in the intervention? (a) Were there parts of the intervention that were helpful? Please give examples if possible. (b) Were there parts of the intervention that did not work out for you? Please give examples if possible.
Did you as parents perceive coming home from the hospital differently? If yes, how? (Only ask this question if there are two parents)	Did you as parents perceive the intervention differently? If yes, how? (Only ask this question if there are two parents)
Was there anything in the neonatal post-discharge process or during the first year at home that you wished were different? If yes, what and how?	Did the intervention have the appropriate scope concerning lengths of visits, number of visits, too much/intensive or too little/low intensity?
Do you have any suggestions for medical healthcare concerning supporting families in the discharge and post-discharge process?	Was there any part of the intervention you wished were different? If yes, which part and how?
Do you want to add anything else?	Do you have any suggestions as to what else could be included in the intervention?
Would it be OK if I called you later if I had any further questions concerning this interview?	Have you seen any effects of the intervention on your child’s behavior or development? If yes, how?

**Table 2 ijerph-17-09326-t002:** Main themes and subthemes of the material.

Main Theme	Subthemes	Dealing with Issues of
1. Child-related concerns	*Continued medical concerns*	Diagnoses and technical supplies.
	*Child regulation*	Questions of eating, sleeping, and digestion.
	*Incomplete recovery when coming home*	The timing of discharge, medical assessment of the child’s state versus the parental perception of the child’s state.
2. Parental inner state	*Loneliness*	From a practical and existential perspective.
	*Ambivalence*	Towards different aspects of the process (i.e., the feeling of relief mixed with worry, or happiness for one’s own child’s development mixed with guilt towards other less fortunate parents).
	*Premature parental identity*	Opposing drives of keeping or letting go of the premature parent label and social media support from peers.
3. Changed family dynamics	*The parental dyad*	Differences in reactions ranging from complementing parental skills to more severe disagreements within the parental dyad was reported in 10 of the 14 families.
	*Thoughts relating to siblings*	(Both in an enriching sense in terms of parenting experience and in a worrisome sense in terms of prioritizing parental attention). In the case of the first child, the absence of siblings was sometimes discussed.
	*Intergenerational support*	When asked about non-hospital related support during the first year at home, grandmothers and grandfathers of the child were mentioned as a main source of help.

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
