# Peer review of "Parents’ Experiences of the First Year at Home with an Infant Born Extremely Preterm with and without Post-Discharge Intervention: Ambivalence, Loneliness, and Relationship Impact"

_ijerph, 2020, doi:10.3390/ijerph17249326_

Round 1
Reviewer 1 Report
Review of:
‘The first year at home with an infant born extremely preterm with and without post-discharge intervention: ambivalence, loneliness and 4 relationship impact’
For the Journal of Environmental Research and Health
Summary:
Thank you for the invitation to review such an interesting paper on a much needed area of research. I felt the methodology matched the topic area and participants and the manuscript overall was quite good. I just have some minor points below.
Title:
- The title is good. Could you put a capital A after the : for Ambivalence?
Abstract
- The abstract was well written and structured.
- Can you define what extremely pre-term in brackets afterwards?
- Are these superordinate themes?
- The abstract is missing a conclusions sentence/s.?
Introduction
- I enjoyed reading the int
- The first sentence in 1.2 does not make sense and point needs to be made clearer. Would an adjustment disorder or comorbid anxiety, depression or PTSD be possible diagnostic terms? Are there additional reasons why you would use qualitative rather than quantitative methods for this topic area?
- Could the first sentence be reordered to say “Over 15 million children worldwide…”?
- Please use brackets around the definition of extremely pre-term.
- Line 71: the sentence here could be better worded.
- Is the point on line 80 mean despite how the child is functioning?
- Once an acronym is defined in the intro please use it thereafter such as WHO.
- The theory of developmental cascade is not given a lot of focus in the introduction but is meant to be the main theory the study is based on. I think more information about the theory could be provided.
Methods
- Could figure one be formatted to look a bit more neat and consistent.
- Can interview be on the one line in the arrow?
- Should it be Table 1 not table 1?
- It may have said it but is there a clear explanation of who transcribed the transcripts and how errors were checked for?
Results
- Table 2 needs full words ‘and’ ‘versus ‘ etc.
- In terms of grandparents in Table 3 do you men grandparents for the child or parents?
- In the results section please make sure what is a superordinate versus a subordinate theme.
- Try and avoid having single sentenced paragraphs throughout the manuscript. Single sentenced paragraphs occurred throughout the manuscript.
- Can you say how many and what percentage of participants mentioned the main themes?
Discussion
- The discussion was good.
- Try and make sure the main theoretical model (the theory of developmental cascade you mentioned in the introduction is clearly discussed in terms of the current studies’ findings.
Author Response
Dear reviewer and editorial board,
Thank you for your elaborated comments, which surely improves our manuscript. In the following sections the comments will be handled in chronological order.
In addition to this we have sent the manuscript for language revision and made all the changes suggested.
|
Reviewer 1 |
|
|
Comment |
Answer |
|
Title |
|
|
The title is good. Could you put a capital A after the : for Ambivalence?
|
Yes, the title is now upgraded to “Parents’ experiences of the first year at home with an infant born extremely preterm with and without post-discharge intervention: Ambivalence, loneliness and relationship impact”. |
|
Abstract |
|
|
Can you define what extremely pre-term in brackets afterwards?
|
The sentence now includes “children born extremely preterm (EPT) before gestational week 28” in line 22-23. |
|
Are these superordinate themes?
|
Yes, there are main themes versus subthemes. We have now made the adjustment that “theme” is changed to “main theme” (which are the superordinate themes) and “subtheme” remains the same.
In addition, there are themes relating specifically to the intervention, which are described as IG-themes.
We have further clarified where to find the answers to “research question I” and “research question II”, in order to separate the main themes and subthemes from the additional IG-themes. |
|
The abstract is missing a conclusions sentence/s? |
Thank you for this vital comment, we added a conclusive line more specifically in line 34-36. Please see the new Abstract for the changed being made, still keeping it under 200 words (currently 195 words). |
|
Introduction |
|
|
The first sentence in 1.2 does not make sense and point needs to be made clearer. Would an adjustment disorder or comorbid anxiety, depression or PTSD be possible diagnostic terms? Are there additional reasons why you would use qualitative rather than quantitative methods for this topic area?
|
Thank you for this clarifying comment. The idea was to separate quantitative from qualitative data. This distinction is more clearly made now, se line 95-96. |
|
Could the first sentence be reordered to say “Over 15 million children worldwide…”? |
Yes, certainly. This is now changed, see line 41. |
|
Please use brackets around the definition of extremely pre-term. |
Yes, please see line 46 for this correction. |
|
Line 71: the sentence here could be better worded.
|
Yes, it contained a typo, thank you for the notification. If you look at line 70-72 the sentence is now divided into two and the meaning is clarified. |
|
Is the point on line 80 mean despite how the child is functioning?
|
Thank you for this interesting food for thought! The answer is not clear in the literature, but the collected clinical experiences from the research team shows that some parents feel emotionally stable even though their child has severe disabilities, while other parents are very affected even by minor child difficulties. In other words, the child’s functioning is not irrelevant, but not the only factor affecting parental emotional state and mental health. |
|
Once an acronym is defined in the intro please use it thereafter such as WHO. |
Thank you! We follow your advice and use the acronyms for WHO and EPT, respectively, after defining them in the Introduction. If the wording “extremely preterm” were mentioned in a citation from an interview, the acronym has not been used for accuracy reasons.
Likewise, the European Standards of Care for Newborn Health is replaced with an acronym (ESCNH) in lines 115 and 189. |
|
The theory of developmental cascade is not given a lot of focus in the introduction but is meant to be the main theory the study is based on. I think more information about the theory could be provided.
|
Thank you for this comment that certainly made us rethink our presentation. Previous research within the field of this study concerns the parental experiences, and the empirical background of this is foremost presented in “1:2 The inner parental experience of preterm birth” and some in “1:1 parental mental health following a preterm birth”.
The main theory of the SPIBI-intervention is the strengths-based perspective, attachment theory, parent-child-interaction and that the parental behavior should be the main target when aiming at affecting child development during the first year. In order to make this clear, some adjustments have been made to the paragraphsdescribing the intervention in general and the “theory of change” in particular (see lines 282-285). |
|
Methods |
|
|
Could figure one be formatted to look a bit more neat and consistent.
|
Yes, certainly. See figure 1 for the new version of the figure. |
|
Can interview be on the one line in the arrow?
|
Yes, certainly. See figure 1 for the new version of the figure. |
|
Should it be Table 1 not table 1?
|
Yes, this is now changed. |
|
It may have said it but is there a clear explanation of who transcribed the transcripts and how errors were checked for? |
The recordings were transcribed verbatim by a research assistant, which may be read in line 342-343. The transcription’s accuracy was facilitated by providing to the transcriber in advance a list of specific terms and acronyms. The content of transcripts was validated by the first author, which had performed the interviews. The second comment has been clarified with a sentence beginning in line 343: “The original recordings were kept, and available to the first author as well as the two authors validating the themes. No errors regarding the content in the transcriptions were detected”. |
|
Results |
|
|
Table 2 needs full words ‘and’ ‘versus ‘ etc.
|
All the “&” are changed to “and”, the vs. is changed to versus, please see Table 2. |
|
In terms of grandparents in Table 3 do you men grandparents for the child or parents?
|
We have no Table 3, but in Table 2 this is correct. Grandparents relates to grandparents of the child. This is changed now (see last paragraph in Table 2). |
|
In the results section please make sure what is a superordinate versus a subordinate theme.
|
Yes, there are main themes versus subthemes. We have now made the adjustment that “theme” is changed to “main theme” (which are the superordinate themes) and “subtheme” remains the same.
In addition, there are themes relating specifically to the intervention, which are described as IG-themes. |
|
Try and avoid having single sentenced paragraphs throughout the manuscript. Single sentenced paragraphs occurred throughout the manuscript.
|
After the restructuring of the revised text, we cannot find any single-sentenced paragraphs. However, there are some paragraphs with only two sentences, that is needed out of structural reasons in order to show the reader the order of main themes – subthemes (answering research question I) and additional intervention-group themes (answering research question II).
Short paragraphs are: 3.1 The first year at home, regardless of group 3.1.1 Main theme 1 3.1.2 Main theme 2 3.1.3 Main theme 3
Our ambition has been to make the structure of the qualitative material and analyzes clear to the reader.
|
|
Can you say how many and what percentage of participants mentioned the main themes?
|
Main themes were essentially omnipresent, since only essential themes were reported in the article. The article also shows the variation and nuances of the answers in the subthemes. In accordance to methodology of thematic analyses, the percentage of different type of answers has intentionally not been reported in the article.
|
|
Discussion |
|
|
Try and make sure the main theoretical model (the theory of developmental cascade you mentioned in the introduction is clearly discussed in terms of the current studies’ findings. |
Thank you for mentioning this, as we now see that the developmental cascade theory may be seen as the main theoretical background of the study, which was not our intention. As far as the discussion is concerned, see changes intended to clarify this in lines 931-934.
The main theory of the SPIBI-intervention is the strengths-based perspective, attachment theory, parent-child-interaction and that the parental behavior should be the main target when aiming at affecting child development during the first year. In order to make this clear, some adjustments have been made to the paragraphsdescribing the intervention in general and the “theory of change” in particular (see lines 282-285). |
Best regards
Erika Baraldi and the research team
Reviewer 2 Report
Manuscript: “The first year at home with an infant born extremely preterm with and without post-discharge intervention: ambivalence, loneliness and 4 relationship impact”
The topic of the manuscript “The first year at home with an infant born extremely preterm with and without post-discharge intervention: ambivalence, loneliness and 4 relationship impact” is very interesting issue for the IJERPH readers. However, the authors need to improve and complete some important questions. Likewise, the content must go through a synthesis process since the current form can be tedious for readers to follow and understand. Some comments are provided in order to help improve the work:
- Title: The study must be identified as a randomised trial in the title.
- Abstract: The abstract of the manuscript is not properly structured and it is skipping some important information that must appear; especially considering that it is an RCT. The authors are recommended to familiarize themselves with the CONSORT-A guide and also to check that their abstract includes the items that are exposed in the CONSORT-A checklist: Hopewell S, Clarke M, Moher D, Wager E, Middleton P, Altman DG, et al. CONSORT for reporting randomized controlled trials in journal and conference abstracts: explanation and elaboration. PLoS Med. 2008;5(1):e20.
3. Introduction:
- The authors shall indicate all their sources of information and some important ones are missing. For example, the origin of data about the incidence of prematurity in Sweden (lines 47-48).
- The introduction is really long and confusing. The must contextualize the objectives of their study in a concise, clear and rational way. Furthermore, it is highly recommended to synthesize the relevant information and eliminate that which does not contribute anything to the study or is not explored in it.
4. Aims: It is advisable that the objectives are formulated as such; in other words, they shall start with an infinitive and being easily measurable by any of the variables collected through the evaluation instruments. Apart from that, the analyzes carried out must be closely linked to them.
5Study design:
- This part includes information about SPIBI that should be explained in the introduction.
- The concept of “corrected age” shall be explained as well as why it is used in premature infants.
- Is this study a pilot study? The sample data provided is not self-explanatory. Have you intended to recruit 130 participants, but finally you only got 17? How the sample size has been determined?
- The authors shall not show results in the methodology. Please, this data should be shown in the first paragraph of results (line 247-252).
- There is a lot of missing information that should be included in the methods of a RCTs part; for example: Description of trial design (such as parallel, factorial) including allocation ratio, important changes to methods after trial commencement (such as eligibility criteria), eligibility criteria for participants, method used to generate the random allocation sequence, type of randomization; details of any restriction (such as blocking and block size), who generated the random allocation sequence, who enrolled participants, and who assigned participants to interventions, blinding ... This must appear in the “procedure” section and must be placed immediately after participants and, before the explanation of the variables and intervention.
- The authors must include a section called “Ethical considerations” and, erase ethical aspects from “Procedure” epigraph.
7. Results:
- The results section must be synthesized since it is very tedious and confusing for the IJERPH reader. Likewise, the results must respond to a specific study objective. The authors are encouraged to review this part in depth.
8. Discussion:
- Again, the authors should synthesize the information provided in the discussion in order to correctly structure it. In the first paragraph, the authors should include the results of their study. Any result that is not mentioned in this first paragraph should not be discussed.
- The results shall be interpreted considering benefits and harms and, taking into account other similar investigations (if any).
- ALL limitations of the RCT must be recognized, addressing sources of potential bias, imprecision ...
- The authors should discuss the generalization of their results based on external validity, as well as clarify their implications.
- The results should be interpreted considering benefits and harms and, taking into account other similar investigations (if any).
9. Conclusions: In this section, it is highly recommended to avoid repeated information that has been already Additionally, they shall limit their conclusions according to the RCTs and their future lines.
10. Other information:
- The authors shall show a registration number and the name of trial registry.
- Where the full trial protocol can be accessed, if available?
Author Response
Dear reviewer and editorial board,
Thank you for your elaborated comments, which surely improves our manuscript. In the following sections the comments will be handled in chronological order.
In addition to this we have sent the manuscript for a language revision and made all the changes suggested.
|
Reviewer 2 |
|
|
Comment |
Answer |
|
The topic of the manuscript “The first year at home with an infant born extremely preterm with and without post-discharge intervention: ambivalence, loneliness and relationship impact” is very interesting issue for the IJERPH readers. However, the authors need to improve and complete some important questions. Likewise, the content must go through a synthesis process since the current form can be tedious for readers to follow and understand. Some comments are provided in order to help improve the work: |
|
|
Title |
|
|
The study must be identified as a randomized trial in the title. |
This study is a qualitative article based on interviews with 17 parents. The study is certainly a part of a larger project, which is defined as an RCT. However, since this article does not represent the RCT, we try to avoid misconceptions by not adding RCT to the title. However, the article is renamed in accordance with reviewer perspectives to ““Parents’ experiences of the first year at home with an infant born extremely preterm with and without post-discharge intervention: Ambivalence, loneliness and relationship impact”. |
|
Abstract |
|
|
The abstract of the manuscript is not properly structured and it is skipping some important information that must appear; especially considering that it is an RCT. The authors are recommended to familiarize themselves with the CONSORT-A guide and also to check that their abstract includes the items that are exposed in the CONSORT-A checklist: Hopewell S, Clarke M, Moher D, Wager E, Middleton P, Altman DG, et al. CONSORT for reporting randomized controlled trials in journal and conference abstracts: explanation and elaboration. PLoS Med. 2008;5(1):e20.
|
Thank you for this advice. Your comments have made us aware that we have not made it sufficiently clear that this qualitative study was not an RCT per se, and we have revised the abstract accordingly.
The RCT is ongoing and recruiting, and hence no final data have been reported. The protocol of the RCT may be found in reference; Baraldi, E.; Allodi, M.W.; Löwing, K.; Smedler, A.C.; Westrup, B.; Ådén, U. Stockholm preterm interaction-based intervention (SPIBI) - study protocol for an RCT of a 12-month parallel-group post-discharge program for extremely preterm infants and their parents. BMC Pediatrics 2020, 20(1), 1-17.
In the protocol, it is stated that all results will be reported in accordance with the CONSORT-agreement and a SPIRIT-checklist is added as supplementary material to the BMC Pediatrics article.
|
|
Introduction |
|
|
· The authors shall indicate all their sources of information and some important ones are missing. For example, the origin of data about the incidence of prematurity in Sweden (lines 47-48).
|
Thank you for your important comment. We added a new reference for the accuracy of our source, Swedish medical birth registry, see ref 3 in the new version of the manuscript. Ref: Cnattingius, S.; Ericson, A.; Gunnarskog, J.; Källén, B. A quality study of a medical birth registry. Scand J Soc Med. 1990, 18(2), 143-148.
The reference is from 1990, but refers to the Swedish Medical Birth Register, which is updated at a state level yearly. The latest update that is available now is from the years 2016-2018 where an average of 362 children were live-born EPT every year.
NB. All references after ref 3 has changed number due to this additional reference to the list. |
|
· The introduction is really long and confusing. The must contextualize the objectives of their study in a concise, clear and rational way. Furthermore, it is highly recommended to synthesize the relevant information and eliminate that which does not contribute anything to the study or is not explored in it. Aims: It is advisable that the objectives are formulated as such; in other words, they shall start with an infinitive and being easily measurable by any of the variables collected through the evaluation instruments. Apart from that, the analyzes carried out must be closely linked to them. |
Thank you for pointing this out to us. We have revised the introduction so that the focus on parental experience is made stronger, and confusion between the present study and the RCT is less likely.
The aims of this qualitative study are: I). How do parents of children born EPT describe the first year at home post discharge? II). How is participating in an interaction- and strength-based home-visiting program perceived by parents of children born EPT? Which we believe are best answered by interviewing the parents about their experiences of the first year at home.
For a full list of the main outcomes of the RCT, we refer to the protocol Baraldi, E.; Allodi, M.W.; Löwing, K.; Smedler, A.C.; Westrup, B.; Ådén, U. Stockholm preterm interaction-based intervention (SPIBI) - study protocol for an RCT of a 12-month parallel-group post-discharge program for extremely preterm infants and their parents. BMC Pediatrics 2020, 20(1), 1-17.
|
|
Study design |
|
|
· This part includes information about SPIBI that should be explained in the introduction.
|
In the “Materials and methods”-section there is a heading 2.4 The intervention. In this section, the SPIBI intervention is briefly described. In this revision, we have added some information concerning the “Theory of change” of SPIBI for clarification (see line 282 and following paragraph).
For full protocol and detailed description of the intervention, we refer to the protocol:
Baraldi, E.; Allodi, M.W.; Löwing, K.; Smedler, A.C.; Westrup, B.; Ådén, U.Stockholm preterm interaction-based intervention (SPIBI) - study protocol for an RCT of a 12-month parallel-group post-discharge program for extremely preterm infants and their parents. BMC Pediatrics 2020, 20(1), 1-17.
However, this manuscript does not report the RCT. |
|
· The concept of “corrected age” shall be explained as well as why it is used in premature infants.
|
OK, thank you for that clarifying comment. In the lines 183-184 and onwards, one sentence has been added: “CA reflects the maturational stage of the child and corresponds to the age the infant would have been if it would have been born at term”.
|
|
· Is this study a pilot study? The sample data provided is not self-explanatory. Have you intended to recruit 130 participants, but finally you only got 17? How the sample size has been determined? |
No, this is not a pilot study, but a qualitative study for which participants have been recruited within a larger project, in which an RCT is conducted with a target of 130 families. The interviews are of 17 parents from 14 of these targeted 130 families.
We have revised the introduction so that the focus on parental experience is made stronger, and confusion between the present study and the RCT is less likely.
In particular the updated Figure 1 and final section of the introduction when presenting the aims, may help to clarify this possible misconception for the readers. |
|
· The authors shall not show results in the methodology. Please, this data should be shown in the first paragraph of results (line 247-252).
|
We have now moved the characteristics of the study subjects to the first section of Results, as suggested. |
|
· There is a lot of missing information that should be included in the methods of a RCTs part; for example: Description of trial design (such as parallel, factorial) including allocation ratio, important changes to methods after trial commencement (such as eligibility criteria), eligibility criteria for participants, method used to generate the random allocation sequence, type of randomization; details of any restriction (such as blocking and block size), who generated the random allocation sequence, who enrolled participants, and who assigned participants to interventions, blinding ... This must appear in the “procedure” section and must be placed immediately after participants and, before the explanation of the variables and intervention.
|
We have now clarified in the introduction on p 226-227 that we did not intend to evaluate the intervention as an RCT. Therefore, we have not included the detailed description here. We refer to the protocol published in BMC Pediatrics, (ref 73) for a full description of all this information.
Baraldi, E.; Allodi, M.W.; Löwing, K.; Smedler, A.C.; Westrup, B.; Ådén, U.Stockholm preterm interaction-based intervention (SPIBI) - study protocol for an RCT of a 12-month parallel-group post-discharge program for extremely preterm infants and their parents. BMC Pediatrics 2020, 20(1), 1-17.
|
|
· The authors must include a section called “Ethical considerations” and, erase ethical aspects from “Procedure” epigraph. · |
Good point, this is added. Thank you for the structural suggestion that really improved our manuscript. (see line 315 and the following paragraph, where text originally written under the “procedure” epigraph was moved).
Moreover, the potential benefits and harms of participating in an interview study have been added to this paragraph (see line 323-328).
|
|
Results |
|
|
· The results section must be synthesized since it is very tedious and confusing for the IJERPH reader. Likewise, the results must respond to a specific study objective. The authors are encouraged to review this part in depth.
|
We have revised the manuscript to make the background, objectives, design and results of this qualitative study clearer.
For increased clarity, we have added which research question is answered by which result in the headings, as follows:
3.1 The first year at home, regardless of group, answering research question I
3.2 Specific themes connected to the intervention, answering research question II
|
|
Discussion |
|
|
Again, the authors should synthesize the information provided in the discussion in order to correctly structure it. In the first paragraph, the authors should include the results of their study. Any result that is not mentioned in this first paragraph should not be discussed. |
We have now revised the first paragraph of the discussion so that it now includes the main findings in relation to the research questions (lines 796-800) and then answering research question I (line 801 and onwards) and after discussing the results of this, the result of research question II is presented (see line 891 and the following discussion of the results). |
|
· The results shall be interpreted considering benefits and harms and, taking into account other similar investigations (if any).
|
This is not an effect study, but a study focusing the subjective experience of parents. However, some possible harms and benefits of being interviewed is added to the new section “ethical considerations” (2.7), see line 323-338. |
|
· ALL limitations of the RCT must be recognized, addressing sources of potential bias, imprecision ...
|
We did not intend to report results of the RCT and hope we have clarified that (see last section in Introduction, lines 226-227). |
|
· The authors should discuss the generalization of their results based on external validity, as well as clarify their implications.
|
Even though this is a qualitative study and hence generalizations may be difficult to make, a paragraph concerning this subject has been added to the discussion of the paper. Please see line 965-974 and onwards to follow the reasoning in this matter.
|
|
· The results should be interpreted considering benefits and harms and, taking into account other similar investigations (if any).
|
Concerning that this is not an RCT, comparing the results of post-discharge interventions programs are not applicable. However, the results of several other studies of parental feelings post-NICU-discharge are presented in 1.2 inner parental experiences of preterm birth. |
|
Conclusion |
|
|
· In this section, it is highly recommended to avoid repeated information that has been already Additionally, they shall limit their conclusions according to the RCTs and their future lines. |
We have revised the conclusions, both in the abstract-section and under the conclusion-headline.
This is not an RCT, why conclusions regarding the RCT is not included in this paper. |
|
Other information |
|
|
· The authors shall show a registration number and the name of trial registry. |
In line 320-322 there is mentioned, and we added the name of the trial as well after this comment. “The study was published in ClinicalTrials.gov in October 2018 (NCT03714633) under the name SPIBI”.
|
|
Where the full trial protocol can be accessed, if available? |
Yes, the full protocol is published in BMC Pediatrics, as mentioned in line 285 in the manuscript (reference 73). Baraldi, E.; Allodi, M.W.; Löwing, K.; Smedler, A.C.; Westrup, B.; Ådén, U. Stockholm preterm interaction-based intervention (SPIBI) - study protocol for an RCT of a 12-month parallel-group post-discharge program for extremely preterm infants and their parents. BMC Pediatrics 2020, 20(1), 1-17.
|
Best regards:
Erika Baraldi and the research team
Reviewer 3 Report
â—‹ Thanks for your effort for working on this paper.
â—‹ This study is meaningful in that it identifies the vivid experiences of parents of premature infants at home during one year after discharge from NICU, and accumulates empirical evidence on the positive effects of home visit programs on these parents.
â—‹ The research problem, purpose, methods, results are well documented. But minor revision needed.
- Adding "Parent's Experience" to the title will make the meaning of this study clear.
- Please describe the RCT process in detail.
- [Figure 1] describes the entire project process. Marking the parts that are directly related to this study will reduce the confusion among readers.
- Line 794-801: Please provide literature evidence and discuss it in more detail.
Author Response
Dear reviewer and editorial board,
Thank you for your elaborated comments, which surely improves our manuscript. In the following sections the comments will be handled in chronological order. In addition to this, we have sent the manuscript to a language check, and made all the suggested changes.
|
Reviewer 3 |
|
|
Comment |
Answer |
|
Adding "Parent's Experience" to the title will make the meaning of this study clear. |
Thank you for this suggestion. The new title is: “Parents’ experiences of the first year at home with an infant born extremely preterm with and without post-discharge intervention: Ambivalence, loneliness and relationship impact”. |
|
Please describe the RCT process in detail |
Thank you for this comment. However, this is not an RCT, but an interview study of 17 parents of extremely preterm born children. If one wants a full description of the RCT, from which these subjects were drawn from, please see;
Baraldi, E.; Allodi, M.W.; Löwing, K.; Smedler, A.C.; Westrup, B.; Ådén, U.Stockholm preterm interaction-based intervention (SPIBI) - study protocol for an RCT of a 12-month parallel-group post-discharge program for extremely preterm infants and their parents. BMC Pediatrics 2020, 20(1), 1-17.
In accordance with this comment, we have revised the Introduction so that the focus on parental experience is made stronger, and confusion between the present study and the RCT is less likely. |
|
[Figure 1] describes the entire project process. Marking the parts that are directly related to this study will reduce the confusion among readers. |
Thank you for this excellent suggestion! Another reviewer wished Figure 1 to be re-made as well and hence all of the suggestions are included in the new figure (see Figure 1). |
|
Line 794-801: Please provide literature evidence and discuss it in more detail. |
Yes, we fully agree that the intergenerational perspective of disabilities and medical diagnoses of grandchildren is an area that should gain more research attention. However, studies of the intergenerational social perspective of prematurity, or even neurodevelopmental psychiatric disabilities are rare, and hence we have re-formulated the referring to this study in Sweden from “previous research” to “a recent study”, see line 813.
|
Best regards:
Erika Baraldi and the research team
Round 2
Reviewer 2 Report
After clarification from the authors, the manuscript is publishable in the current format according to my point of view.